# FASTER GRADIENT DESCENT IN DEEP LINEAR NETWORKS: THE ADVANTAGE OF DEPTH

## ABSTRACT

Gradient descent dynamics in deep linear networks has been studied under a wide range of settings. These studies have reported some *negative results on the role of depth*, in that, gradient descent in deep linear networks: (i) can take exponential number of iterations to converge, (ii) can exhibit sigmoidal learning, i.e., almost no learning in initial phase followed by rapid learning, (iii) can delay convergence with increase in depth. Some of these results are also under stronger assumptions such as *whitened data* and *balanced initialisation*. These messages from prior works suggest that *depth hurts the speed of convergence*.

In this paper, we argue that the negative role of depth in the prior works is due to certain pitfalls which can be carefully avoided. We give a *positive message on the role of depth*, i.e., seen as a additional resource, depth can always be used to speed up convergence. For this purpose, we consider scalar regression with quadratic loss. In this setting, we propose a novel *aligned gradient descent* (AGD) algorithm for which we show that (i) linear convergence is always possible (ii) *depth accelerates the speed of convergence*. In AGD, feature alignment happens in first layer and the deeper layers accelerate by learning the right scale. We show acceleration in AGD happens in finite time for *unwhitened* data. We provide insights into the acceleration mechanism and also show that acceleration happens in phases. We also demonstrate the acceleration due to AGD on synthetic and benchmark datasets. Our main message is not to propose AGD as a new algorithm in itself, but to demonstrate that *depth is an advantage* in linear networks thereby dispelling some of the past negative results on the role of depth.

## 1 INTRODUCTION

Deep learning has been successful in a wide variety of machine learning tasks. There are two key ingredients to this success namely (i) as the depth increases, deep models can *express* a rich class of non-linear functions and (ii) most often that not, the right function from this rich class can be found by gradient descent, a simple algorithm. As a result, analysing the dynamics of gradient descent in deep models has turned out to be an important question in machine learning. However, this analysis is quite challenging due to the presence of non-linearity.

Deep linear networks have been widely studied in the literature Arora et al. (2018a;b); Bartlett et al. (2018); Ji & Telgarsky (2018); Ziyin et al.; Saxe et al. (2014); Atanasov et al. (2021), mainly as an analytical surrogate for deep non-linear networks. In the case of linear networks, increasing depth does not help in increasing the expressivity. However, the dynamics of gradient descent is non-linear, and exhibits similar phenomena observed in deep non-linear networks. At the same time, due to the absence of non-linear activation functions, the analysis of deep linear networks turn out to be simpler than the non-linear counterparts.

Analysis of gradient descent dynamics in deep linear models also is a steep task. Some of the results are under restrictive assumptions such as *whitened data* Saxe et al. (2014); Arora et al. (2018a); Bartlett et al. (2018) and *balanced initialisation* Arora et al. (2018b); Atanasov et al. (2021) and *small learning rate* Saxe et al. (2014); Arora et al. (2018b). Even under such restrictions, the results in the literature paint a rather *negative* picture of of the role of depth. To elaborate, it has been observed that gradient descent in deep linear networks: (i) can take exponential number of iterations to converge Shamir (2018), (ii) can exhibit sigmoidal learning, i.e., almost no learning in initial

phase followed by rapid learning Saxe et al. (2014), (iii) can delay convergence with increase in depth Saxe et al. (2014); Arora et al. (2018b). Given that such phenomena are empirically observed in deep non-linear networks as well, the overall belief is that *depth hurts convergence even in linear networks*.

In this paper, we decouple the analysis of gradient descent in deep linear networks from its utility in serving as an analytical surrogate for deep non-linear networks. Instead, our aim is to understand the role of depth when compared to one layer shallow linear networks. While increasing depth in linear networks increases the computational overhead, the question is whether it is possible speed up the convergence. In other words, we investigate the extra computations vs faster convergence trade-off. For the problem of scalar linear regression with quadratic loss, under minimal assumptions on the data, we ask and answer the following question:

*Can a deep linear network achieve faster convergence in finite time than a shallow one layer linear network? If so, what is the extra computational overhead?*

**Contributions:** In this paper, we provide a *positive answer* to the above question, that is, depth *helps in increasing* the speed of convergence. Our specific contributions are

- We propose a novel algorithm called *aligned gradient descent* (AGD) (Algorithm 1) for deep linear networks and prove (Theorem 1) that it achieves faster convergence in *finite time* than gradient descent on a shallow one layer linear network.

- AGD provides instance-wise acceleration, in that, for any learning learning rate that is stable for the shallow linear network, AGD achieves faster convergence in deep linear network for the same learning rate.

- We show that AGD requires only $5L$ extra computations (per iteration and per example) when compared to one layer shallow linear network. This is only a negligible increase in the computations when the number of input dimension is much larger than $L$.

- We provide insights into the mechanism underlying acceleration and demonstrate empirically that AGD achieves acceleration in phases one for each eigenvector component.

- We demonstrate via numerical experiments on synthetic and standard datasets that our AGD on deep linear networks indeed converges faster than the one layer shallow linear network.

**Organisation:** In Section 2, we present the setting and the basic convergence result for gradient descent in one layer shallow linear networks. In Section 3, we discuss briefly the negative results on the role of depth reported in prior works. We point out the possible pitfalls which cause such negative results. In Section 4, we present our aligned gradient descent algorithm, present its finite time convergence result and discuss how it avoids the pitfalls in the prior works. In Section 4, we also discuss the acceleration mechanism in AGD (Section 4.3), demonstrate its instance-wise speed up property and study its role of depth (Section 4.2), and demonstrate that it achieves faster convergence (in comparison to GD in shallow network) on standard datasets (Section 4.5). In Section 5 we review the related works and make concluding remarks in Section 6.

**Our Message:** While AGD indeed speeds up convergence, our goal is far from selling AGD as a better optimisation method for scalar linear regression. Rather, via AGD our aim is to demonstrate that some of the *apparently negative* results on the role of depth are not an inherent property of depth but are due to some pitfalls which are carefully avoided in AGD. In other words, our goal is to show that depth as a resource works to our advantage in linear networks, and AGD is one of the ways to achieve it.

## 2 Setup and Gradient Descent in Shallow Linear Network

In this section, we present the setup, assumptions, and state the finite time analysis of gradient descent for shallow linear network (Proposition 1).

**Notation:** We use $[m]$ to denote the set $\{1, \ldots, m\}$. We use $\Theta = \{\Theta^{(l)}, l \in [L]\} \in \mathbb{R}^{d_{\text{net}}}$, where $\Theta^{(l)} \in \mathbb{R}^{d_{l-1} \times d_l}$ are weights of layer $l \in [L]$. $\Theta^{(-l)} = \{\Theta^{(i)} : i \in [L], i \neq l\}$ denotes the weights of all layers other than layer $l$. $\nabla_{\Theta}$ and $\nabla_{\Theta^{(l)}}$ respectively denote the gradient with respect to weights

of the entire network and the gradient with respect to the weights of layer $l$. For $n \times n$ matrices $B_1, \ldots, B_t$, $\prod_{s=1}^{t} B_s = B_t B_{t-1} \cdots B_1$. We use $\Theta^{(l:L)} = \prod_{l'=l}^{L} \Theta^{(l')}$ to denote the effective weight matrix from layer $l$ to $L$.

**Assumption 1** *For the dataset* $(x_i, y_i)_{i=1}^{n} \in \mathbb{R}^d \times \mathbb{R}$*, let* $X = [x_1, \ldots, x_n] \in \mathbb{R}^{d \times n}$ *be the data matrix . We assume (i)* ***Bounded Labels:*** $|y_i| \leq 1, \forall i \in [n]$ *and (ii)* ***Full Rank:*** *Rank(X)=*$\min\{n, d\}$.

**Linear Networks:** We consider linear networks of depth $L$ and $d_l$ hidden units in each layer with scalar output. When $L = 1$, we call it a ***shallow one layer linear network***, and for $L > 1$ we call it a ***deep linear network***. The input dimension is $d_0 = d$, and output dimension is $d_L = 1$. For the input feature vector $x \in \mathbb{R}^d$, the hidden layer outputs $x^{(l)}, l \in [L]$ are given by: $x_{\Theta}^{(0)} = x, x_{\Theta}^{(l)} = \Theta^{(l)} x_{\Theta}^{(l-1)}$ and the final output is given by $\hat{y}(x; \Theta) = x_{\Theta}^{(L)} = \Theta^{(L)} \cdots \Theta^{(1)} x$. We consider quadratic loss which is given by $\mathcal{L}(\Theta) = \frac{1}{n} \sum_{i=1}^{n} \frac{1}{2} (\hat{y}(x_i; \Theta) - y_i)^2 = \frac{1}{n} \sum_{i=1}^{n} \frac{1}{2} (\Theta^{(L)} \cdots \Theta^{(1)} x_i - y_i)^2$.

**Definition 1** *Let* $\lambda_i \in \mathbb{R}^n$ *be the eigenvalues of a* $n \times n$ *real symmetric positive semi-definite matrix* $A$*. Define* $\lambda_{\min}^{eff}(A) = \min\limits_{i \in [n]: \lambda_i \neq 0} \lambda_i$ *as the minimum eigenvalue which is greater than* $0$*. For the matrix* $I - A$*, define the spectral radius to be* $\rho(I - A) \stackrel{def}{=} \max\limits_{i \in [n]} |1 - \lambda_i|$*, and the effective spectral radius to be* $\rho^{eff}(I - A) \stackrel{def}{=} \max\limits_{i \in [n]: \lambda_i \neq 0} |1 - \lambda_i|$*.*

**Definition 2** *Let* $Y_* = (y_i, i \in [n]) \in \mathbb{R}^n$ *and* $\widehat{Y}_{\Theta} = (\hat{y}(x_i; \Theta), i \in [n]) \in \mathbb{R}^n$ *be the vectors of true and predicted outputs. For* $n > d$*, define* $Y_*^{sol} \stackrel{def}{=} X^{\top} (X X^{\top})^{-1} X Y_*$*, and for* $n \leq d$*, define* $Y_*^{sol} \stackrel{def}{=} Y_*$*. Define the error vectors* $E_{\Theta} \stackrel{def}{=} Y_* - \widehat{Y}_{\Theta}$ *and* $E_{\Theta}^{sol} \stackrel{def}{=} Y_*^{sol} - \widehat{Y}_{\Theta}$*.*

**Proposition 1** *For gradient descent in shallow one layer linear network with fixed constant learning rate of* $\eta > 0$*, we have*

$$E_{\Theta_{t+1}} = \left( I - \frac{\eta}{n} X^{\top} X \right) E_{\Theta_t} \tag{1}$$

$$\left\| E_{\Theta_{t+1}}^{sol} \right\|_2 \leq \rho^{eff} \left( I - \frac{\eta}{n} X^{\top} X \right) \left\| E_{\Theta_t}^{sol} \right\|_2 \tag{2}$$

**Proof:** Please refer to any standard textbook on optimisation Boyd & Vandenberghe (2004).

**Corollary 1** *If* $\rho^{eff}(I - \frac{\eta}{n} X^{\top} X) < 1$*, then*

$$\|\widehat{Y}_{\Theta_t}\|_2 < 2\|Y_*\|_2$$

**Proof:** $\|\widehat{Y}_t\|_2 \leq \|E_t^{sol}\|_2 + \|Y_*^{sol}\|_2 < \|E_0^{sol}\|_2 + \|Y_*^{sol}\|_2 = 2\|Y_*^{sol}\|_2 \leq 2\|Y_*\|_2$.

**Choice of Learning Rate:** If we know $\lambda_{\max}(X^{\top} X)$ and $\lambda_{\min}^{eff}(X^{\top} X)$, then the optimal learning rate is $\eta_* = \frac{2}{\lambda_{\max}(X^{\top} X) + \lambda_{\min}^{eff}(X^{\top} X)}$. However, when such knowledge is absent, we can find $\eta$ via hyper-parameter tuning to ensure $\rho^{eff} \left( I - \frac{\eta}{n} X^{\top} X \right) < 1$, i.e., to ensure stable convergence.

**Effective spectral radius** $\rho^{eff}$ is used in the case when $n > d$ and the matrix $X^{\top} X$ has $n - d$ zero eigenvalues. Thus, even for the optimal choice of learning rate $\rho(I - \frac{\eta}{n} X^{\top} X) = 1$ due to the $n - d$ zero eigenvalues of $X^{\top} X$. However, $\rho^{eff}$ captures the convergence to $Y_*^{sol}$.

## 3 PRIOR WORK : ROLE OF DEPTH IN LINEAR NETWORKS

Understanding gradient descent (GD) dynamics in deep non-linear networks is an important problem in machine learning. However, given the difficulty of this problem, many works have studied gradient descent dynamics on deep linear networks for the following reasons:

- Even though the output is a linear function of the input, the dynamics of gradient descent in deep linear networks is still non-linear.

- Deep linear networks exhibit many phenomena also found in deep non-linear networks such as long plateaus followed by fast transition to lower error solutions Saxe et al. (2014).

- In linear networks increasing depth does not alter their expressivity. Hence the role of depth in optimisation can be clearly separated Arora et al. (2018b).

Even analysing gradient descent in deep linear networks in the most general setting is a steep task. Thus, many works make several restrictive assumptions such as *small learning rates* Arora et al. (2018a;b), *whitened data* Arora et al. (2018a), *balanced initialisation* Arora et al. (2018b); Ji & Telgarsky (2018); Atanasov et al. (2021), *near identity initialisation* Bartlett et al. (2018), $l_p$ *losses for* $p > 2$ Arora et al. (2018b) to provide some useful insights. Such restrictive assumptions for linear networks are not a deterrent as long as one can replicate and obtain insights into phenomena occurring in deep non-linear counterparts. That said, some of the messages in the past works apparently seem to suggest that *depth hurts speed of convergence* even in deep linear networks. In what follows, we review some of those results.

### 3.1 PLATEAUED CONVERGENCE: NUMBER OF ITERATIONS SCALES EXPONENTIAL IN DEPTH

Shamir (2018) considered the gradient descent dynamics in one-dimensional (i.e., $d_l = 1, \forall l \in [L]$) networks, i.e., the problem of learning $\hat{y}(x; \Theta) = \Theta^{(L)} \cdots \Theta^{(1)} x$ where the weights are all scalars ($\forall l \in [L], \Theta^{(l)} \in \mathbb{R}$). It was shown that for small enough learning rate, for Xavier (**Theorem 2**,Shamir (2018)) and near-identity initialisation (**Theorem 3**, Shamir (2018)) for gradient descent to converge, the number of iterations required *scales exponentially in depth*. It was demonstrated *empirically* that such phenomena also occur in multi-dimensional networks. The essence of their argument is captured in the following example.

Consider a two layer network with one hidden unit whose output is given by $\hat{y}(x) = \Theta(2)\Theta(1)x$. The dataset has only one data point, which is $x = 1, y = 1$. The loss function is $\mathcal{L}(\Theta) = (\Theta(2)\Theta(1) - 1)^2$. In this example, for $\Theta_0(1) = 1, \Theta_0(2) = -1$ and infinitesimally small learning rates, it can be shown via simple calculations that, $\forall t \geq 0, \Theta_t(1) = -\Theta_t(2), \Theta_t(1) > 0, \Theta_t(2) < 0$, and $\Theta_t(1) \to 0_+$ and $\Theta_t(2) \to 0_-$, i.e., the output $\hat{y}(x) \to 0_-$. For finite but small learning rates, the loss hits a *plateau* and stays there for exponentially large number of iterations.

**Possible Pitfall and Our Fix:** The key issue here is that at initialisation itself the network output is aligned in the wrong direction with respect to the target. We fix this issue in our aligned gradient descent algorithm (Algorithm 1), by ensuring that in iteration, the alignment of the output of the deep linear network with the target is as good as the alignment in a shallow one layer network.

### 3.2 SIGMOIDAL CONVERGENCE : DEPTH DELAYS CONVERGENCE

Saxe et al. (2014) showed that deep linear networks incur a delay in convergence when compared to shallow linear networks. Saxe et al. (2014) also considered the one-dimensional problem of learning $\hat{y}(x; \Theta) = \Theta^{(L)} \cdots \Theta^{(1)} x$, in the dataset is given by $x = 1, y_1 = y_*$. The output of the network is given by $\hat{y}_t = \Theta_t^{(L)} \cdots \Theta_t^{(1)}$, and the loss is given by $\mathcal{L}(\Theta_t) = \frac{1}{2}(y_* - \hat{y}_t)^2$. To simply the analysis they considered a small initialisation for all the weights i.e., $\forall l \in [L], \Theta_0^{(l)} = \epsilon' > 0$. Let $e_t = y_* - y_t$, then for an infinitesimally small learning rate, the error dynamics in continuous time can be given by $\dot{e}_t = -L(y_t)^{2(\frac{L-1}{L})} e_t$. However, in discrete time we have to choose a small but finite learning rate which results in stable dynamics. This learning rate is shown decay with depth as $O(\frac{1}{Ly_*^2})$. To see why this causes a delay in convergence as depth increases, we can set $y_* = 1, y_0 = \epsilon, \eta = \frac{\eta'}{L}$. Let $e_t^{(L)}$ be the error dynamics of a network of depth $L$. The approximate (discrete time) progress in the first iteration is given by

$$e_1^{(L)} = (1 - \eta' \epsilon^{2(\frac{L-1}{L})}) e_0^{(L)} \tag{3}$$

While it is true that as $t \to \infty$, $e_t^{(L)} \to 0$, from the fact that $\epsilon^{2(\frac{L-1}{L})}$ decreases as $L \to \infty$, it is also follows that $e_1^{(1)} < e_1^{(2)} \ldots < e_1^{(l)}$ and hence $e_t^{(1)} < e_t^{(2)} \ldots < e_t^{(l)}$ i.e., the error at any time $t$ is smaller in the shallow networks. In other words, increasing the depth delays the speed of convergence.

**Possible Pitfall and Our Fix:** The key issue here is managing scale of the output during training, in that, $O(\frac{1}{Ly_*^2})$ is too conservative. We fix this issue in our aligned gradient descent algorithm (Algorithm 1), in two ways: (i) we initialise the first layer weights to be all 0 and the weights of the rest of the layers to be 1 (this takes care of the scale at $t = 0$), (ii) we use adaptive learning rates that are based on the growth of the weights (this takes care of the managing the scale during training, i.e., $t > 0$).

### 3.3 Depth speeds up convergence for $l_p$ loss only when $p > 2$ and not for $p = 2$

Arora et al. (2018b) show that gradient descent in deep linear network is equivalent to gradient descent in shallow linear network with a preconditioning scheme. It is argued that depth acts like momentum with adaptive learning rates, in that, increasing depth promotes movement along directions already taken by optimisation. In particular, Arora et al. (2018b) argue that in shallow networks have no "communication" between the weights which can hurt the optimisation in case of $l_p$ loss (for $p \in 2\mathbb{N}, p > 2$). For this, they consider the example dataset $x_1 = (1, 0)^\top, x_2 = (0, 1)^\top$ and labels $y_1, y_2$. For a shallow network $\hat{y}(x; \Theta) = x(1)\Theta(1) + x(2)\Theta(2)$ and the loss is given by $\mathcal{L}^p(\Theta) = \frac{1}{p}(\Theta(1) - y_1)^p + \frac{1}{p}(\Theta(1) - y_2)^p$. Let $e_t(i) = y_i - \hat{y}(x_i; \Theta_t)$ be the error whose dynamics for small learning rate $\eta > 0$ is given by $e_{t+1}(i) \leftarrow (1 - \eta(e_t(i))^{p-2})e_t(i)$. Say $\Theta_0(1), \Theta_0(2)$ are initialised close to 0, then $e_0(1) \approx -y_1$ and $e_0(2) \approx -y_2$. So for stable convergence we need $\eta < \frac{2}{y_i^{p-2}}$. Now, for $p = 2$ the $y_i^{p-2}$ term in denominator has no role to play in the choice of the learning rate. However, when $p > 2$, the learning rate is determined by $\max\{y_1, y_2\}$ and the least of the two coordinates converges slowly. It is argued that deep linear networks can alleviate this issue, in that, since their weights are tied to one another, there is communication between them, and such communication results in an adaptive preconditioning.

Arora et al. (2018b) also observe in their experiments on a linear regression problem with quadratic loss deeper networks converge slowly than shallow ones.

**Possible Pitfall and Our Fix:** The key issues are (i) only dynamics of the weights is analysed and not the features, (ii) for the quadratic case, it is argued that a problem dependent information cannot be used in the choice of the learning rate. We fix this in our aligned gradient descent algorithm (Algorithm 1) by explicitly controlling the features learnt which in turn helps us to choose an adaptive learning rates.

## 4 Our Work : Aligned Gradient Descent

In this section, we propose *aligned gradient descent* (AGD) Algorithm 1 which achieves a linear rate of convergence. In AGD, all the hidden units are aligned to the same feature direction; key idea is to let the first layer to align to the right feature direction and the deeper layers accelerate in the feature direction aligned in the first layer. In order to keep the learning stable, AGD also uses adaptive learning rates which are implicitly derived from the iterates of the algorithm itself. In what follows, we present AGD and describe its key ingredients and state its finite time convergence result in Theorem 1.

**The key ingredients of AGD in Algorithm 1** are

• **Width is 1:** This reduces the computational overhead. We argue in Appendix B in comparison to GD on one layer shallow network, AGD requires only $5L$ extra multiplication and $L$ extra addition operations.

• **Initialisation, Scale and Alignment:** We also initialise the first layer weights to be all 0 and the rest of the layer weights to be 1. The first layer *aligns*, i.e., learns the one-dimensional feature $\Theta^{(1)}x$ and the rest of the layers just scale, i.e., $x^l = \Theta^{(l)}x^{(l-1)}$ (since width is 1, for $l = 2$ to $L$, $\Theta^{(l)}$ are all scalars). Thus the output of the deep linear network is aligned with respect to the target at the very first gradient step itself and this alignment continues during the entire course of training.

• **Adaptive Learning Rates:** Let $K_\Theta^{(l)} \overset{\text{def}}{=} \sum_{i,j} x^{(l-1)}(j) \times \Theta^{(l+1:L)}(1, i)$ denote the feature Gram matrix, i.e., kernel matrix corresponding to layer $l$. It can be easily shown that $K_{\Theta_t}^{(1)} =$

---

**Algorithm 1** Aligned Gradient Descent

Initialise : $\Theta_0^{(1)} = \mathbf{0}_{1 \times d}, \forall l = 2, \dots, L, \Theta_0^{(l)} = 1$
**for** $t = 0, 1, \dots, T$ **do**
$\quad \Theta_{t+\frac{1}{L}}^{(1)} \leftarrow \Theta_t^{(1)} - \frac{\eta}{\left(\Theta_t^{(2:L)}\right)^2} \nabla_{\Theta^{(1)}} \mathcal{L}(\Theta_t)$
$\quad \Theta_{t+\frac{1}{L}}^{(-1)} \leftarrow \Theta_t^{(-1)}$
$\quad$**for** $l = 2, \dots, L$ **do**
$\qquad \Theta_{t+\frac{l}{L}}^{(l)} \leftarrow \Theta_{t+\frac{l-1}{L}}^{(l)} - \frac{1}{2}\left(\Theta_{t+\frac{l-1}{L}}^{(l)}\right)^2 \nabla_{\Theta^{(l)}} \mathcal{L}(\Theta_t)$
$\qquad \Theta_{t+\frac{l}{L}}^{(-l)} \leftarrow \Theta_{t+\frac{l-1}{L}}^{(-l)}$
$\quad$**end for**
**end for**
**return** $\Theta_T$

---

$\left(\Theta_t^{(2:L)}\right)^2 \left(X^\top X\right)$ and $K_{\Theta_{t+\frac{l-1}{L}}^{(l)}} = \frac{1}{\left(\Theta_{t+\frac{l-1}{L}}^{(l)}\right)^2} (\widehat{Y}_{\Theta_{t+\frac{l-1}{L}}})(\widehat{Y}_{\Theta_{t+\frac{l-1}{L}}})^\top$. Thus the updates in Algorithm 1 can be seen to choose adaptive learning rates of $\eta_t^{(1)} = \frac{\eta}{\left(\Theta_t^{(2:L)}\right)^2}$ in the first layer, and

$\eta_t^{(l)} = \frac{1}{2}\left(\Theta_{t+\frac{l-1}{L}}^{(l)}\right)^2$ for layers $l = 2$ to $L$.

• **Round Robin Updates:** Algorithm 1 updates only one layer at a time. Thus, each of the update is equivalent to learning in a one layer shallow network with the hidden features of that layer. This enables us to invoke Equation (1) after updating each layer (see Equations (4) and (6)).

**Theorem 1** *In Algorithm 1, let $\eta > 0$ be any stable learning rate for gradient descent in one layer shallow network. It follows that (we drop $\Theta$ in $\Theta_t$ for the sake of clarity)*

$$E_{t+\frac{1}{L}} = \left(I - \frac{\eta}{n} X^\top X\right) E_t \tag{4}$$

$$\left\| E_{t+\frac{1}{L}}^{sol} \right\|_2 \le \rho^{eff}\left(I - \frac{\eta}{n} X^\top X\right) \left\| E_t^{sol} \right\|_2 \tag{5}$$

$$E_{t+\frac{l}{L}} = \left(I - \frac{1}{2n}(\widehat{Y}_{t+\frac{l-1}{L}})(\widehat{Y}_{t+\frac{l-1}{L}})^\top\right) E_{t+\frac{l-1}{L}} \tag{6}$$

$$\left\| E_{t+\frac{l}{L}}^{sol} \right\|_2 < \left\| E_{t+\frac{l-1}{L}}^{sol} \right\|_2 \tag{7}$$

$$\left\| E_{t+1}^{sol} \right\|_2 < \rho^{eff}\left(I - \frac{\eta}{n} X^\top X\right) \left\| E_t^{sol} \right\|_2 \tag{8}$$

**Proof:** We argue the under-parameterised ($n < d$) and over-parameterised ($n \ge d$) cases separately.

• **Convergence in under parameterised regime:** For $n > d$, let $\Theta_* = (XX^\top)^{-1} X Y_*$ be the unique solution. Now, $Y_*^{sol} = X^\top \Theta_* \in \mathbb{R}^n$ and $\widehat{Y}_\Theta = X^\top \Theta$ lies in the column span of the matrix $X^\top$. Thus Equations (4) and (5) are same as Equations (1) and (2) and AGD makes as much progress as gradient descent does in a one layer shallow network. The update in layers $l = 2$ to $L$ (Equation (6)) only reduces the error in the component of $\widehat{Y}_{t+\frac{l-1}{L}}$. From Corollary 1, it follows that $\left\| \widehat{Y}_{t+\frac{l-1}{L}} \right\|_2^2 < 2n$, and hence $\rho^{eff}\left(I - \frac{1}{2n}(\widehat{Y}_{t+\frac{l-1}{L}})(\widehat{Y}_{t+\frac{l-1}{L}})^\top\right) < 1$. Now, since $\widehat{Y}_{t+\frac{l-1}{L}}$ also lies in the column span of $X^\top$ we have Equation (7). Combining everything we have Equation (8). Thus, $E_t^{sol} \to 0$, and since $X$ is full rank, we can infer that $\Theta_t^{(1:L)} \to \Theta_*$.

**Convergence in over parameterised regime:** For $n \le d$, let $\Theta_* \in \mathbb{R}^d$ be the least norm solution which is the unique solution which belongs to the subspace spanned by the features, i.e., $\Theta_* \in \text{Span}(\{x_i, i \in [n]\})$. Since the matrix $X^\top X$ is full rank $E_t \to 0$. Further, the first layer weights

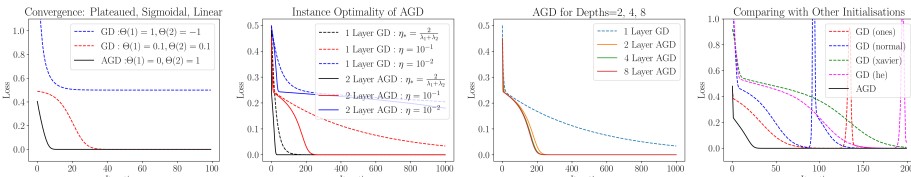

Figure 1: In the left most plot, the setting is the same as the one described in Section 3.1 and the algorithms are run with $\eta = 0.1$. Here, the blue and red dotted line corresponds to plateaued and sigmoidal convergence respectively. The black solid line corresponds to AGD. The explanation for the second, third and fourth plots (from the left) can be found in the text in Section 4.2.

$\Theta_t^{(1)}$ are linear combination of the features $x_i \in \mathbb{R}^n$ and since the weights of layers $l = 2$ to $L$ are scalars, the effective weight $\Theta_t^{(1:L)} \in \text{Span}(\{x_i, i \in [n]\}), \forall t \geq 0$. Thus $\Theta_t^{(1:L)} \to \Theta_*$.

### 4.1 DISCUSSION OF ALGORITHM 1 AND THEOREM 1

**Avoiding the pitfalls in prior work and achieving linear rate:** The round robin updates ensure that at every iteration AGD aligns its output to the target as well as a gradient descent update in a one layer shallow network. This avoids the plateauing phenomena reported in Shamir (2018). Further, by properly bookkeeping the scales, AGD uses adaptive learning rates which ensure that the depth dependent delay phenomena of Saxe et al. (2014) is avoided. These are demonstrated in left most plot of Figure 1. Further, Arora et al. (2018b) considered *whitened data* and argued that the communication between the weights in a deep linear network helps faster convergence. However, we show (see Section 4.4) that for *unwhitened* data such communication can actually cause unstable behaviour the gradient descent dynamics. We also show (see Section 4.4) how our AGD avoids such unstable behaviour.

**Acceleration in AGD:** The rank-one correction in the direction of the predicted output vector as show in Equation (6) speeds up the convergence of AGD when compared to GD on one layer shallow network. The mechanism of acceleration is explained in Section 4.3. Also, the speed up in AGD is instance-wise, i.e., for any learning rate $\eta > 0$ which is stable for gradient descent in one layer network, AGD for the same $\eta$ speeds up the convergence of the deep linear counterpart. This instance-wise speed up, the effect of depth and comparison with GD in deep linear networks with other initialisation scheme is discussed in Section 4.2

### 4.2 INSTANCE-WISE SPEED, EFFECT OF DEPTH, OTHER INITIALISATION

In this subsection, we consider a dataset with $d = 2$ and $n = 2$, where $x_1 = (\sqrt{2}, 0)^\top$ and $x_2 = (0, \sqrt{2 \times 0.01})^\top$ and $y_1 = 1, y_2 = 1$.

**Instance-wise speed up** is shown in Figure 1 (the second from left plot) for three different learning rates namely $\eta_1 = \eta_* = \frac{2}{1+0.01}, \eta_2 = 10^{-1}, \eta_3 = 10^{-3}$. The GD runs are shown in dotted lines and the AGD runs (for the same learning rates) on a 2 layer network are shown in solid lines. Here, AGD converges to zero loss faster than the corresponding GD runs for all the three instances $\eta_1, \eta_2, \eta_3$. Note that $\eta_1 = \eta_*$ is the optimal learning rate for GD in a shallow one layer network and AGD outperforms it, i.e., *the best AGD in a deep networks is better than the best GD in a shallow one layer linear network.*

**Effect of Depth:** For $\eta = 0.1$, the performance of deep linear networks of depth $L = 2, 4, 8$ trained with AGD and one layer shallow linear network trained with GD are shown in Figure 1 (the second plot from right). We observe that the advantage of depth diminishes as depth increases. This is because, layers 2 to $L$ can speed up learning only in the output direction and the advantage of such speed up diminishes in the later layers if the initial layers itself learn the right scale.

**Comparison with other initialisation:** We compared the performance of AGD and GD with standard initialisation schemes for two layer network. For each of these schemes, we tuned for the best learning rate, and for AGD we chose $\eta_* = \frac{2}{1+0.01}$. The results are shown in the right most plot of

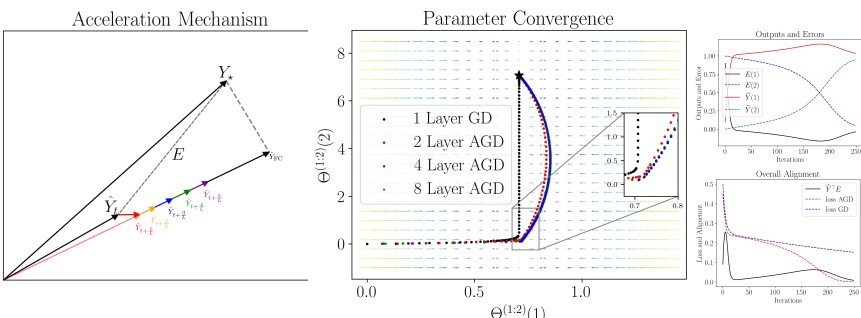

Figure 2: The left most cartoon illustrates how AGD speeds up convergence in its layers $2$ to $L$. The middle plot shows the effective parameter $\Theta_t^{(1:2)}(1)$ and $\Theta_t^{(1:2)}(2)$. It can be seen that the accelerated trajectory takes a curved path in comparison to the GD trajectory. The right most plots show the alignment of the output and error to explain their role in acceleration in AGD.

Figure 1. The GD runs are shown in dotted lines and AGD in solid line. We see that AGD performs better than GD with standard initialisation. Further, we see from the plots that for the normal and He initialisations, gradient descent has unstable behaviour, in that, the loss reduces first and then it overshoots and again converges to zero. This unstable behaviour of GD in deep networks is explained in Section 4.4.

### 4.3    ACCELERATION IN AGD : MECHANISM AND PHASES (FIGURE 2)

We continue with the same dataset described in Section 4.2 and the results are in Figure 2.

**Acceleration Through Layers (left most cartoon in Figure 2):** At time $t$, layer $1$ update corrects the output as $\widehat{Y}_{t+\frac{1}{L}} = \widehat{Y}_t + \frac{\eta}{n}(X^\top X)E_t$. Subsequent layers $l = 2, \ldots, L$ correct the output as

$$
\widehat{Y}_{t+\frac{l}{L}} = \widehat{Y}_{t+\frac{l-1}{L}} + \frac{1}{2n}\left((\widehat{Y}_{t+\frac{l-1}{L}})(\widehat{Y}_{t+\frac{l-1}{L}})^\top\right)E_{t+\frac{l-1}{L}} = \left(1 + \frac{1}{2n}\langle\widehat{Y}_{t+\frac{l-1}{L}}, E_{t+\frac{l-1}{L}}\rangle\right)\widehat{Y}_{t+\frac{l-1}{L}}
$$

In other words, layers $l = 2, \ldots, L$ progressively accelerate the output in the same direction. For the case of $L = 2$ and $\eta_t^{(2)} = \frac{1}{\|\widehat{Y}\|_2^2}$ ensures *full correction*, i.e., $\widehat{Y}_{t+1}$ is the *projection* of $Y_*$ in the direction given by $\widehat{Y}_{t+\frac{1}{2}}$.

**Acceleration Phases Through Time (right most plots in top and bottom in Figure 2:** For acceleration to happen due to layers $l = 2, \ldots, L$, we need the factor $|\langle\widehat{Y}_{t+\frac{l-1}{L}}, E_{t+\frac{l-1}{L}}\rangle|$ to be significant i.e., vectors $\widehat{Y}_{t+\frac{l-1}{L}}$ and $E_{t+\frac{l-1}{L}}$ have to be aligned sufficiently. As the learning progresses, the network output $\widehat{Y}$ learns components in eigenvector directions corresponding to the increasing order of $|1 - \eta\lambda_i|$. At the same time, the error due to these components diminish in the error term $E = Y_* - \widehat{Y}$. In the *whitened* case, since all $|1 - \eta\lambda_i|$ are same, there is only one acceleration phase. However, in the *unwhitened* case, depending on the spread of the $|1 - \eta\lambda_i|$ the acceleration manifests as phases; utmost one phase each for each eigenvalue. Since the *unwhitened* case is more general we will illustrate it via the following example.

Consider AGD in a two layer network with $1$ hidden unit, and the same dataset as in Section 4.2. Here, $\frac{1}{n}X^\top X = \mathbf{diag}(\lambda_1, \lambda_2)$, where $\lambda_1 = 1$ and $\lambda_2 = 0.01$. In the first phase of $\widehat{Y}(2) \approx 0$ and the network learns $\widehat{Y}(1) \uparrow Y_*(1)$, and hence $\widehat{E}(1) \downarrow 0$ (see right most top plot in Figure 2 from iterations $0$ to $50$). Thus in the first phase, the quantity $\langle\widehat{Y}, E\rangle \approx \widehat{Y}(1)E(1)$ is first small, then increases, and then decreases again (see right most bottom plot in Figure 2 from iterations $0$ to $50$). In the next phase (iterations $50$ to $250$), $\widehat{Y}(2) \uparrow Y_*(2)$, and $E(2) \downarrow 0$, and in this phase $E(1) \approx 0$. Similar to the previous phase, the quantity $\langle\widehat{Y}, E\rangle \approx \widehat{Y}(2)E(2)$ is small, then increases in the middle of the phase and then decreases.

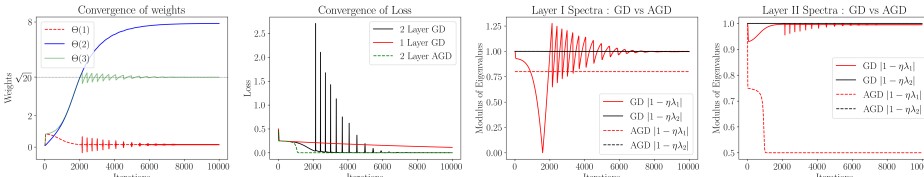

Figure 3: These plots that demonstrate that gradient descent in deep networks for unwhitened data can have unstable behaviour. It is also shown here that AGD does not suffer from such unstable behaviour. Please refer to the text in Section 4.4 for an explantion of these plots.

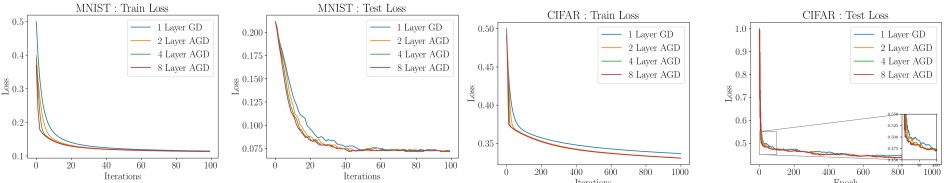

Figure 4: Shows the performance of AGD for various depths in MNIST and CIFAR. Please see Section 4.5 for a description of the plots.

## 4.4 STABILITY ON UNWHITENED DATA : AGD VS GD

**AGD Algorithm 1 avoids unstable ($\rho^{\text{eff}} > 1$) behaviour:** Prior works have looked at gradient descent dynamics in deep linear networks with the assumption that the data is *whitened* Saxe et al. (2014); Arora et al. (2018a). We show that violating this assumption might cause undesirable convergence behaviour of the gradient descent even in a two layer network for reasonably small learning rate. We consider the network whose output given by $\hat{y}(x;\Theta) = \Theta(1)\Theta(3)x(1) + \Theta(2)\Theta(3)x(2)$ (here, $\Theta(1),\Theta(2)$ are first layer weights and $\Theta(3)$ is the second layer weight). For this network, we consider the dataset with $n = 2$, $d = 2$ and $x_1 = (\sqrt{2},0)^\top, x_2 = (0,\sqrt{2}\times 0.02)^\top$ and $y_1 = 1, y_2 = 1$, and we set $\eta = 0.1$. Here, $\frac{1}{n}X^\top X = \begin{bmatrix} \lambda_1 & 0 \\ 0 & \lambda_2 \end{bmatrix}, \lambda_1 = 1, \lambda_2 = 4\times 10^{-4}$ and $I - \frac{\eta}{n}K^{(1)} = \begin{bmatrix} 1 - \eta\lambda_1\Theta(3)^2 & 0 \\ 0 & 1 - \eta\lambda_2\Theta(3)^2 \end{bmatrix}, I - \frac{\eta}{n}K^{(2)} = \begin{bmatrix} 1 - \eta\lambda_1\Theta(1)^2 & -\eta\sqrt{\lambda_1\lambda_2}\Theta(1)\Theta(2) \\ -\eta\sqrt{\lambda_1\lambda_2}\Theta(1)\Theta(2) & 1 - \eta\lambda_2\Theta(2)^2 \end{bmatrix}$. To minimise the loss, as $t \to \infty$ we need $\Theta(1)\Theta(3) \to \frac{1}{\sqrt{2}}$ and $\Theta(2)\Theta(3) \to \frac{50}{\sqrt{2}}$. $\hat{y}_1 = \sqrt{2}\Theta(1)\Theta(3)$ being associated with the largest eigenvalue is learnt first. This happens by iteration 40 as shown in Figure 3. Now, in the second phase, the network learns $\hat{y}_2 = \sqrt{2}\Theta(2)\Theta(3)$, which causes the magnitude of $\Theta(3)$ to increase. Note that for convergence to be stable, we need $|1 - \eta\lambda_1\Theta(3)^2| < 1$, i.e., $\Theta(3) < \sqrt{20}$. However, once $\Theta_t(3) > \sqrt{20}$, the network starts exhibiting unstable behaviour. AGD (Algorithm 1) avoids this by using $\eta^{(1)} = \frac{\eta}{\Theta(3)^2}$. As shown in Figure 2 (second plot from right), the loss profile of GD (black solid line) has too many spikes due to the unstable behaviour. In contrast, the AGD loss profile (green dotted line) is smooth. We have also plotted the GD in one layer for the same dataset for ease of comparison. The third and the fourth plots show that the spectrum of the matrices $\left(I - \frac{\eta}{n}K_{\Theta_t}^{(1)}\right)$ and $\left(I - \frac{\eta}{n}K_{\Theta_t}^{(2)}\right)$ (dotted lines for AGD and solid lines for GD). As discusses above, in the case of GD, the spectrum of the said matrices grow in magnitude (greater than 1) which causes the overall unstable behaviour. However, in the case of AGD, the spectrum never grows in magnitude greater than 1.

## 4.5 AGD VS GD ON MNIST AND CIFAR-10

We compared GD in shallow one layer network with AGD in deep linear network on standard datasets namely MNIST and CIFAR-10. Since we consider only the scalar regression setting, for MNIST we chose only classes $\{3,8\}$ and we labelled them $-1$ and $+1$ respectively. Similarly, for CIFAR-10 we chose only $\{bird, airplanes\}$ and labelled them $-1$ and $+1$ respectively. For both

datasets, we first tuned for the best learning rates for GD in shallow one layer network, i.e., we found $\eta_{*,\text{GD}}^{\text{MNIST}}$ and $\eta_{*,\text{GD}}^{\text{CIFAR}}$. We then used these learning rates to run AGD on deep linear networks of depth $2, 4, 8$. For both datasets, AGD performs better than GD in train as well as test data (Figure 4).

## 5 RELATED WORK

The literature on deep linear network (DLN) is rich, and of these, works relevant to us are of two kinds (i) those on gradient descent and (ii) those on neural tangent kernel (NTK) alignment.

Saxe et al. (2014) analysed GD with infinitesimal learning rate. They showed that DLNs too exhibit phenomena found in deep non-linear networks such as plateauing followed by fast transition to low error solutions. They also showed that deep linear networks incur a delay in learning speed relative to shallow networks. However, they provided a class of random orthogonal initialisation under which this delay in learning speed is finite even as the depth of network approached infinity. Arora et al. (2018b) studied the advantage of depth in DLNs because in DLNs depth cannot help in expressiveness and if any can only can help in optimisation. They showed that in the scalar regression setting with $l_p$ loss ($p > 2$), under a *balanced initialisation of weights*, depth helps in acceleration. They showed that this acceleration due to depth cannot be achieved by other preconditioning schemes. However, in their experiments they also reported that in the $l_2$ case, depth *mildly hurts* the speed of convergence. Bartlett et al. (2018) analyse a gradient descent for quadratic loss and *whitened* data. They consider a specific sub-class, in that, they consider only *linear residual networks* with uniform width across layers. For this setting, they show that gradient descent takes polynomial (in the depth) iterations to converge an $\epsilon$ approximate solution. Arora et al. (2018b) extended Bartlett et al. (2018) significantly and showed that for $l_2$ over *whitened* data and *balanced weight initialisation*, gradient descent converges at linear rate. Shamir (2018) prove (under mild assumptions) for one-dimensional network that for standard random initialisation, the number of iterations required for convergence scales exponentially with the depth. Atanasov et al. (2021) look at the dynamics of the neural tangent kernel during training of deep non-linear and linear networks for the case of quadratic loss, under *small and balanced weight initialisation* and *whitened* data. In the case of DLNs, the analytically establish the phenomena of *silent alignment* wherein, in the early stage (phase I) of training the NTK aligns (to the final NTK) and during rest of the training (phase II) the NTK evolves only in scale. This implies that the learnt function equivalent to a kernel regression solution with the final NTK. They also show that this alignment kernel depends on the depth. However, they also demonstrate that non-whitened data weakens the silent alignment effect.

**Our work:** Saxe et al. (2014); Arora et al. (2018b); Shamir (2018) note that depth *slows* down the rate of convergence. On the contrary, we show that depth actually *speeds* up convergence. In comparison to Arora et al. (2018a); Bartlett et al. (2018), we do not make any *whitening* assumption and our results hold for any general data matrix. Our initialisation scheme also differs from the aforementioned works which either use randomised or balanced or near identity initialisation. Another important aspect of our results is that it is finite time. Atanasov et al. (2021) only analyse alignment in the case of whitened data and do not use alignment in an algorithm. In contrast, in our work, we use alignment (even for unwhitened data) to propose an algorithm which speeds up convergence.

## 6 CONCLUSION

Gradient descent (GD) in deep linear networks have been widely studied in literature mainly as a surrogate to deep non-linear networks. The aim of such study is to throw light on phenomena that occur in deep non-linear networks. Prior works on deep linear networks have reported that depth plays a negative role in the convergence of gradient descent in deep linear networks. In this paper, we argued that such negative results are due to pitfalls which we carefully avoid in our novel aligned gradient descent algorithm (AGD). We presented finite time guarantees for AGD and showed that it achieves accelerated convergence when compared to GD on a shallow one layer network in theory as well as experiments on synthetic and benchmark datasets. We discussed the acceleration mechanism of AGD. We also show pointed out how GD for deep linear networks can lead to unstable convergence when data is unwhitened, and we showed that AGD does not suffer from such unstable behaviour. We conclude that depth is an advantage and helps to speed up convergence in deep linear networks.

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

| Op. | $\eta_t^{(1)}\nabla_{\Theta_t^{(1)}}\mathcal{L}$ | $\Theta_{t+\frac{1}{L}}^{(1)}$ | $x_{t+\frac{1}{L}}^{(1)}$ | $\hat{y}_{t+\frac{1}{L}}$ | $\eta_t^{(l)}\nabla_{\Theta^{(l)}}\mathcal{L}$ | $\Theta_{t+\frac{l}{L}}^{(l)}$ | $x_{t+\frac{l}{L}}^{(l)}$ | $\hat{y}_{t+\frac{l}{L}}$ | $\Theta_{t+1}^{(l:L)}$ | Total* |
|---|---|---|---|---|---|---|---|---|---|---|
| Mul | $2d$ | $0$ | $d$ | $1$ | $3$ | $0$ | $1$ | $1$ | $L-1$ | $3d+5L$ |
| Add | $d$ | $d$ | $(d-1)$ | $0$ | $0$ | $1$ | $0$ | $0$ | $0$ | $3d+L$ |

Table 1: Addition and Multiplication operations in AGD for the simpler $d_{\text{hid}} = 1$ architecture. Here, $l = 2, \ldots, L$. Total* is an upper bound for the total number of operations.

## A  APPENDIX

## B  EXTRA COMPUTATIONS DUE TO ROUND ROBIN UPDATES IN AGD

Note that, due to our initialisation, at $t = 0$, the forward pass computations are trivial, i.e., $\hat{y}(x) = 0$, and $\Theta^{(l:L)} = 1, \forall l = 2, \ldots, L$. Further, as we proceed through $t + \frac{l}{L}$ for $l = 1, \ldots, L$, at each step we have to perform forward pass ($\hat{y}_{t+\frac{l}{L}}$) backward pass ($\eta_t^{(l)}\nabla_{\Theta^{(l)}}\mathcal{L}$) and weight update ($\Theta_{t+\frac{l}{L}}^{(l)}$) — these have been accounted for in Table 1. The total computations are upper bounded by $3d + 5L$ multiplications and $3d + L$ additions.

In a shallow one layer network, we need (i) multiplications: $d$ in the forward pass and $2d$ in the backward pass, (ii) additions: $d - 1$ in the forward pass, $d$ for backward pass and $d$ for weight update. Thus, the total computations are $3d$ multiplications and $3d - 1$ additions.

**Depth Overhead:** From the above arguments, it is clear that computational overhead of going deep is upper bounded by $5L$ multiplications and $L$ additions per example and per update (i.e., from $t$ to $t + 1$ when updating all layers).

