# OpenReview forum: "Faster Gradient Descent in Deep Linear Networks: The Advantage of Depth"
_ICLR.cc/2025/Conference — Submitted to ICLR 2025_

### Official Review · Reviewer_joZK · 2024-10-22

**Soundness:** 3
**Presentation:** 3
**Contribution:** 2
**Rating:** 3
**Confidence:** 4

**Summary:**

Whether the depth provides an advantage in training deep networks has been a key question discussed in prior research. In response to this, the present study constructs a gradient method (along with the initialization strategy) that achieves fast convergence even as the number of layers increases, positively answering this question. The distinct feature compared to previous studies is that un-whitened data is allowed, as well as the use of the standard L2 loss function.

**Strengths:**

- The authors constructed a concrete and solvable example where a specific gradient method achieves fast convergence in a deep linear network. Notably, this work demonstrates that depth can be an advantage in optimization even in the general case with un-whitened data.

- The intuition behind the speed-up of convergence is well explained, including examples from previous works on acceleration.

**Weaknesses:**

**Limitation on the Width**
As far as I understand, the analysis is limited to a width of 1 for hidden layers, and it is not obvious whether it can be extended to networks with general widths. If this is true, it is a very restricted scenario.

**Necessity of AGD**
While it is clear that AGD contributes to the acceleration of convergence, the necessity of using AGD specifically is unclear. For instance, why wouldn’t Adam or Newton’s method work?

**Questions:**

**Limitation on the Width**
- Why do the authors not generalize the results to networks with general widths? A width of 1 is very restrictive. The authors mention some prior work as being limited due to the assumption of whitened data, but to me, restricting the width to 1 seems more idealized and far removed from modern *over-parameterized* deep networks.
- What is the width used in the experiments on datasets in Section 4.5?

**Necessity of AGD**
AGD has an adaptive learning rate scaled by $(\Theta_t^{(2:L)})^2$.  Is it necessary to use the square, or can this be generalized to $(\Theta_t^{(2:L)})^p$ ($p>0$)?

---

> ### Author Response · Authors · 2024-11-16
> **Many thanks to Reviewer joZK; Our point by point response**
>
> We thank the reviewer for the review. We now provide a point by point response
>
> **Weakness 1**: Our algorithm can be extended in a straightforward manner to general widths and we present the same below
>
>
> -------------------------
>
> Aligned Gradient Descent for general width $w$
>
> ------------------------
> Initialise : $\Theta^{(1)}_0 =\mathbf{0},\forall l=2, \ldots, L, \Theta^{(L)}_0=\mathbf{1}$
>
> for $t = 0,1,\ldots T$ do
>
>  $\eta_t^{(1)}\leftarrow \frac{\eta}{\Theta_t^{(2:L)}(1,1)^2\times w}$
>
>  $\Theta^{(1)}_{t+\frac1L}\leftarrow \Theta^{(1)}_t-\eta^{(1)}_t \nabla_1 \mathcal{L}(\Theta_t)$
>
>  $\Theta^{(-1)}_{t+\frac1L}\leftarrow\Theta^{(-1)}_t$
>
>   for $l=2,\ldots, L$ do
>
>  $\eta_t^{(1)}$ $\leftarrow \frac{1}{2}(\Theta^{(1)}_{t+\frac {l-1}L}(1,1))^2 \times w^2$
>
>   $\Theta_{t+\frac lL}^{(l)}$ $\leftarrow \Theta^{(l)}_{t+\frac {l-1}L}-\eta^{(l)}_t\nabla_l \mathcal{L}(\Theta_t)$
>
>  $\Theta_{t+\frac lL}^{(-l)}\leftarrow\Theta^{(-l)}_{t+\frac {l-1}L}$
>
> Due to the nature of the initialisation,  all the weights in a given layer are identical throughout training. Thus, the gradient dynamics of AGD for width $w>1$ and $w=1$ are identical.
>
>
> **Weakness 2** The question related to Adam and Newton is tangential in the context. To elaborate,
>
> $\bullet$ Consider convergence speed as a function $f$(depth, optimisation-algorithm), we say that if we fix optimisation-algorithm = GD (AGD is gradient descent with adaptive learning rates), we want to understand $f$(depth, GD) as a **function of depth**. **In our paper**, we show that $f$ is monotonically better with depth.
>
> $\bullet$ The reviewer's question makes sense when only when one studies $f$(depth/architecture, optimisation-algorithm) by fixing depth/architecture, and then looks at the behaviour of $f$(fixed-depth/architecture, optimisation-algorithm) as a **function of optimisation-algorithm**. **In this context (which is not the objective of our paper), we agree with the reviewer**, for a fixed depth/architecture algorithms such as Adam and Newton are better.
>
>
>
>
> --------------------------
>
> Answer to Questions
>
> 1. We are not restricted to width = 1, and we have presented AGD for general width $w>1$.
>
> 2. We used width =1 in Section 4.5.  Due to the nature of the initialisation,  all the weights in a given layer are identical throughout training. Thus, the gradient dynamics of AGD for width $w>1$ and $w=1$ are identical.
>
>
> 3. Yes it is necessary to use $(\Theta_t^{(2:L)})^2$, and it cannot be generalised to $(\Theta_t^{(2:L)})^p$ for $p>0$, doing so may cause instability in the updates.

---

### Official Review · Reviewer_REzX · 2024-10-29

**Soundness:** 1
**Presentation:** 2
**Contribution:** 1
**Rating:** 1
**Confidence:** 4

**Summary:**

The paper tackles the optimization issues induced by higher depth.

The authors show that in the case of 1) width 1, 2) a very precisel and unrealistic initialization where all the layers but one are initialized at 1 and the others at 0, a version of GD converges exponentially as the standard OLS.

The authors conclude that this implies that some negative optimization results attributed to the depth are not actually attributable to the depth.

**Strengths:**

The paper is clearly written and well organized. The problem is well introduced.

**Weaknesses:**

The paper does not meet the ICLR standards for theoretical novelty or practical relevance. The issues with unrealistic assumptions, limited scope of the investigation, and lack of empirical support are significant enough that a strong reject recommendation is given.
The paper has a huge number of problems, but those are not even the point. It completely fails in its objective. The conclusions drawn are misleading and not supported by the computations on the toy models provided.

The claims of non-detrimental effects of depth are based on a single, highly contrived example where all layers but one are essentially bypassed by setting their weights to 1, reducing the model to a standard linear regression. This approach does not reflect the complexities or realities of deploying deep networks in practice. The authors provide precisely one of those few cases (of measure zero) in which depth is not detrimental for the optimization and they claim that thus depth may not be detrimental.

This does not imply that deeper networks may be trainable. They are actually cooking up an example in which they kill the effect of depth to say that depth there has no effect. Not only this cooked up example is very far from practice, but everywhere else in the parameter space, depth has an effect.

It is also misleading to suggest that the computational increase is limited to 5L, which results solely from choosing a network width of one—a characteristic of the architecture, not the algorithm itself. Computational demands typically scale with network width.

Moreover, even in this overly simplistic case they show that GD is unstable and that is the reason why they change algorithms. In practice they are telling us that even on this instance of deep network in which all the layers are the identity except one, GD would not behave well as the depth scales.

Additionally, the terminology used to describe shallow networks as standard linear regressions contradicts the literature referenced, where shallow networks are generally recognized as having a single hidden layer.

Overall, this paper does little to enhance our understanding of the phenomena it intends to explore and fails to address its central thesis effectively.

A minor point of critique is the redundancy in the text; the last paragraph of page 1 is essentially repeated with identical wording at line 168 and again in section 5. This redundancy could have been avoided to streamline the content and enhance clarity.

**Questions:**

Am I wrong about my assessment? I really do not see how this analysis can lead to prove your claims but happy to discuss it with the authors.

---

> ### Author Response · Authors · 2024-11-15
> **Many thanks to reviewer REzX (in this comment, we point out the factual errors/misconceptions in the review)**
>
> **We are very thankful for the detailed review and also to the reviewer's openness for a discussion**. The review makes points which are useful as a devil’s advocate. However, there are critical factual errors and misconceptions in the review which make the overall assessment incorrect. We first point out these errors and misconceptions, then provide a point by point response to the review, and then present our questions to the reviewer.
>
> **In what follows, to make our position clear, we have indicated where and why we disagree. That said, at the end of the day, we find the points raised by Reviewer REzx very valuable in making the contribution stronger and we are very thankful for the same.**
>
> ---------------------
> ----------------------
>
> **Disagreement with the Summary** : The summary is also incorrect due to the factual inaccuracies which we point out below.
>
> **Factual Error 1 in the Review  (multiple points on all the layers are the identity except one; reducing it to standard linear regression)**: We strongly disagree. We are training all the layers, so for $t>0$ none of the layers are identity. So, our models are not equivalent to standard linear regression with one layer for time $t>0$. In particular, the dynamics of gradient descent in our models is not the same as one layer model.
>
> **Factual Error 2 in the Review  (point on GD is unstable)**: We strongly **disagree**. The failure case, i.e., unstable behaviour of GD is also for other standard (in the market) initialisation schemes namely He and Normal under a small constant learning rate. This has been already mentioned explicitly in the title of Section 4.1, and in boldface text which reads “comparison with other initialisations”. The instability of GD is further explained in Section 4.4 where the analysis does not make use of any specific initialization (i.e., all layers identity except one).
>
> **Factual Error 3 in the Review  (depth killed to a point of no effect)** : We **disagree**. Since we are training all the layers, depth has a huge effect in the gradient dynamics, and if not addressed properly (like we do in AGD), depth can cause instability.
>
> **Misconception 1 in the Review  ( point on contrived example; overly simplistic case)**: We strongly **disagree**. We would like to re-emphasise that the negative results on depth in prior works (Saxe et al. (2013), Arora (2018), Shamir (2019)) are essentially arguments based on width = 1 networks. Here, we (us and prior works) are studying the phenomenology of depth and width = 1 is sufficient for the phenomena to emerge.
>
> $\bullet$ Saxe et al (2013) (https://arxiv.org/pdf/1312.6120) use connectivity modes, and reduce the model to width = 1. We quote lines from page 7 above equation (14) of their work “each connectivity mode in the $N_l$ layered network can be described by $N_{l − 1}$ scalars $a^1 , . . . , a^{N_{l} −1}$.
>
> $\bullet$ Shamir (2019) (http://proceedings.mlr.press/v99/shamir19a/shamir19a.pdf) studies only scalar networks. We quote the first line of the abstract "We study the dynamics of gradient descent on objective functions of the form $f(\prod^k_{𝑖=1}w_i)$ (with respect to scalar parameters $w_1,…,w_k$)".
>
> $\bullet$ Arora et al (2018) (https://proceedings.mlr.press/v80/arora18a/arora18a.pdf) note in Section 8 (page 7, column 1, paragraph 2) “Specifically, it shows that in the evaluated set-ting, hidden layers of size 1 (scalars) suffice in order for the essence of overparameterization to fully emerge”, i.e., they are considering the width =1 case.  Further, they also go on to add, we quote from (page 7, column 2, paragraph 1, line 1) “As can be seen, convergence of deeper networks is (slightly) slower in the case of l2 loss. This falls in line with the findings of Saxe et al. (2013)”.
>
> **Misconception 2 in the Review  (measure zero)** : Considering deep linear networks with arbitrary initialisation as a class, we pick the right member (not an average representative) that achieves our objective of using depth to increase the speed of convergence.
>
> **Misconception 3 in the review  (this does not imply that deeper networks may be trainable)** : We have not made any claims for deep non-linear networks at all in our paper. If the reviewer meant deep linear networks, then we strongly **disagree**. We are indeed showing that deep linear networks are trainable, and even a stronger claim that they are instance-wise (dataset and learning rate) faster than one layer networks.

---

> ### Author Response · Authors · 2024-11-15
> **Point by Point Response to other comments by reviewer REzx**
>
> **Limited scope of the investigation/practical relevance** : We **partially agree**. Our work does not immediately translate into a practice recommendation for deep non-linear networks.  However, from prior results, it is natural to conjecture that even for deep linear networks depth hurts, i.e., no deep linear network can outperform a one layer linear network in speed of convergence. Our work invalidates such a conjecture, i.e., negative role of depth is not a settled case, and the hope is that in the future one can come up with initialisations and algorithms such that depth helps even optimisation in deep non-linear networks.
>
> **Failed Objective** : We **disagree**. Our objective is to use depth as a resource to increase the speed of convergence in deep linear networks when compared to one layer model. Our work indeed achieves this objective.
>
> **Lack of Theoretical novelty** : We **disagree**. AGD is a novel algorithm. There are two important novelties, the initialisation and the adaptive scaling both of which contribute towards the increased speed of deep linear networks in comparison to one layer network.
>
> **Lack of empirical support**: We strongly **disagree**. Subsections 4.1, 4.2, 4.3, 4.4, 4.5 and Figures 1 to 4 are entirely empirical in nature and support the theory and the objective.
>
> **Unrealistic Initialisation**: We **disagree**. One can call our initialisation non-standard and not widely used in practice. Such widely used realistic initialisations are not useful for our objective. This is the very reason we came up with our initialisation.
>
> **Unrealistic Assumption**: We strongly **disagree**. If at all, we are making very minimal assumptions on the dataset. We have already explained initialisation and the width = 1 aspects in the previous points.
>
> **No enhancement of our understanding of the phenomena**: We **disagree**. The paper clearly separates the issue of poor initialisation and the possible unstable behaviours that occur due the training dynamics. In fact, ours is the first work to demonstrate how unwhitened data causes GD to be unstable even for two layer width 1 linear networks.
>
> **Computational increase**: We **agree**. The increase limited to 5L is only due to architecture. We will modify this in our revision.
>
> **Terminology used to describe shallow networks**: Point taken. We will change this in our revision.
>
> **Redundancy**: Point taken. We will change this in our revision.

---

> ### Author Response · Authors · 2024-11-15
> **Questions to Reviewer REzx**
>
> 1) All prior works for deep linear networks (in venues comparable to ICLR)  have essentially considered width = 1. Yet, they not only failed to improve speed of convergence using depth, but also gave a message (based on a rather heuristic gradient flow analysis) that depth hurts. We have also considered width = 1, and showed (in theory and experiments) that depth helps to speed up convergence in finite time. Why does this not amount to ICLR standards for theoretical novelty? What is our fault here? doing the right thing to fix an issue?
>
> 2) In optimising any objective in a continuous domain, a unique global solution is a single point whose measure is zero. The non-trivial part is to actually find this measure zero global solution. Our objective is to use depth to increase the speed of convergence and we indeed find such a solution for the objective. What is our fault here? finding a solution for the objective?
>
> 3) Nowhere in the paper we are talking about deep non-linear networks. Further, we have clarified in the introduction (lines 58-63 in our paper) that our objective is to use depth to increase speed of convergence in deep linear networks. We wonder then why our work is being held against what it may or may not imply for deep non-linear networks. Does the reviewer consider every work on deep linear networks unrealistic because no one uses them in practice?

---

> > ### Comment · Reviewer_REzX · 2024-11-27
> >
> > Thanks for the comments. Sorry but I believe there is no space for improving my grade of strong rejection. Some general comments:
> > 1.  *All prior works* seems a little strong statement, can you point me to these works?
> > 2. I'm sorry I didn't properly convey my comment about measure zero. It was not about solutions of the objective. It was about the fact that I worked a lot with linear networks and when analyzing wider-deeper linear networks, I believe this behavior you describe is never the case in practice. It may be the case, but with probability zero over initialization. It is the case that increasing the depth in linear networks training becomes longer in general, even if you initialize them to the identity.
> > Good luck with your future submissions.

---

> ### Author Response · Authors · 2024-11-27
> **Clarification on "All the prior works"; Depth does not help in practice**
>
> **All prior works is a little stronger: Agreed** (sorry for the same). We meant **all prior works which we discussed in our paper** and which we again mentioned in our response above. These are **Saxe et al (2013)** (https://arxiv.org/pdf/1312.6120), **Shamir (2019)** (http://proceedings.mlr.press/v99/shamir19a/shamir19a.pdf), **Arora et al (2018)** (https://proceedings.mlr.press/v80/arora18a/arora18a.pdf). Especially, **Arora et al (2018)** has these lines "**Specifically, it shows that in the evaluated setting, hidden layers of size 1 (scalars) suffice in order for the essence of overparameterization to fully emerge**".
>
>  Listing these priors works and reposting part of the response we wrote.
>
> $\bullet$ **Saxe et al (2013)** (https://arxiv.org/pdf/1312.6120) use connectivity modes, and reduce the model to width = 1. We quote lines from page 7 above equation (14) of their work “each connectivity mode in the $N_l$ layered network can be described by $N_{l − 1}$ scalars $a^1 , . . . , a^{N_{l} −1}$.
>
> $\bullet$ **Shamir (2019)** (http://proceedings.mlr.press/v99/shamir19a/shamir19a.pdf) studies only scalar networks. We quote the first line of the abstract "We study the dynamics of gradient descent on objective functions of the form $f(\prod^k_{𝑖=1}w_i)$ (with respect to scalar parameters $w_1,…,w_k$)".
>
> $\bullet$ **Arora et al (2018)** (https://proceedings.mlr.press/v80/arora18a/arora18a.pdf) note in Section 8 (page 7, column 1, paragraph 2) “Specifically, it shows that in the evaluated set-ting, hidden layers of size 1 (scalars) suffice in order for the essence of overparameterization to fully emerge”, i.e., they are considering the width =1 case.  Further, they also go on to add, we quote from (page 7, column 2, paragraph 1, line 1) “As can be seen, convergence of deeper networks is (slightly) slower in the case of l2 loss. This falls in line with the findings of Saxe et al. (2013)”.
>
>
> **Not observed in practice**: We **agree** with your comment on your experience about what happens in **current practice**. Implicitly **we are questioning the current practice**. What we are showing is a positive result of depth under the specific initialisation scheme (which is not identity initialisation) and adaptive learning rate. In other words, if we want to use depth to our advantage, what we are suggesting is the **good practice** and other schemes that do not increase speed with depth is **bad practice** (we are not sure why one should be bothered about bad practice).

---

> > ### Comment · Reviewer_REzX · 2024-12-02
> >
> > I advise for next time you submit this work to work out mathematically better the difference between your model and theirs!
> > For instance you could frame your result in those settings if it is applicable, it is not immediately obvious for me.
> > Good luck!

---

### Official Review · Reviewer_XUVi · 2024-11-04

**Soundness:** 2
**Presentation:** 2
**Contribution:** 1
**Rating:** 3
**Confidence:** 4

**Summary:**

This work studies optimization in deep linear networks, in particular the effect of depth on convergence.
It proposes a novel algorithm, aligned gradient descent (AGD), that solves the issue of slow convergence of linear networks.

**Strengths:**

The results seem correct.

**Weaknesses:**

This work considers a very narrow problem that, in my opinion, is of very little interest.
First of all, the problem of deep linear networks is very narrow.
However, even worse than that, the authors motivate their work from trivial observations.
Section 3.1 describes a trivial situation in which the neural network is initialized at a very special value of the parameters, that is known to converge to a saddle point. Any initialization that is sufficiently far from that special case would not suffer from the limitations described by the authors. Also, depth plays no role in this section, contrary to what seems to be the main motivation of the authors.
Similarly, section 3.2 describes another trivial situation where the neural network is initialized very near the saddle point. Again, any initialization that is far away enough from that special initialization would not suffer from the problems described.
In section 3.3, the authors desribe problems in the case of p-norm loss, but that is also a narrow case of very little interest.
The novel algorithm, AGD, is quite complicated and is limited to deep linear networks.
It remains unclear why that algoroithm may be useful or interesting in any other (non-linear) case.

**Questions:**

NA

---

> ### Author Response · Authors · 2024-11-16
> **Effectively the review is empty; Our point to point response**
>
> We thank the reviewer. Please find below our point to point response to the reviewer comments:
>
> 1. **(Deep linear networks is very narrow)** This is a sweeping opinion on the entire line of research on deep linear networks. We do not know how to respond to this and neither we believe our response is going to change the reviewers opinion.
>
> 2. **Section 3.1, 3.2, 3.3** The points raised by the reviewer are issues with prior work which provide fundamental insights about the role of depth. There are some factual errors in the review and we point out them below:
>
> $\bullet$ **Depth indeed plays in Section 3.1** Shamir (2019) (http://proceedings.mlr.press/v99/shamir19a/shamir19a.pdf), the main result Theorem 2 reads as follows
> "The following holds for some positive constants c, c′ independent of k: Under Assumptions 1 and 2, if gradient descent is ran with any step size $ \eta ≤ exp(ck)$, then with probability at least $1 − exp(−\Omega(k))$ over the initialization, the number of iterations required to reach suboptimality less than $c′$ is at least $exp(\Omega(k))$", where $k$ is the depth.
>
> $\bullet$ **Section 3.2 is not trivial** In Section 3.2, we have paraphrased Section 2 of Saxe et al. (2013) (https://arxiv.org/pdf/1312.6120). The weight initialisation considered is any small initialisation around origin (not particularly vanishingly small) and is quite general.
>
> $\bullet$ **Section 3.3 $p$-norm** The case of $p$-norm considered by Arora et al. (2018) is significant in that, it was first work in the literature to show that depth helps in optimisation. Downplaying such a critical result by saying that it is very narrow misses the point entirely.
>
> 3. **AGD is complicated** : It would have been more useful if the reviewer provided a justification of why AGD is complicated. That said, AGD is gradient descent with adaptive learning rates and we believe is quite simple and elegant.
>
> 4. We too agree. For now it AGD is not immediately applicable to deep non-linear networks. We have mentioned the same in our abstract as well as introduction. That said, however result dispels the negative role of depth in case of linear networks. We hope this is a good starting point for future works to use depth to their advantage in optimising deep non-linear networks as well.
>
>
> **Disappointing Review**: We are disappointed to find that **the review is effectively empty because**
>
> $\bullet$ First two lines express a sweeping negative opinion about why deep linear networks are uninteresting.
>
> $\bullet$ Points made about sections 3.1, 3.2, 3.3 are about prior work and not about our work.
>
> $\bullet$ The only comment about our work is that AGD is complicated and no reasons are given.
>
>
> **Question to the reviewer: If the reviewer felt so negatively about deep linear networks, why choose to review the paper in the first place?** The reviewer could have declined to review the paper. Keeping aside the review scores for now, we would have at least obtained a useful review which helps to forward a discussion and strengthen our contribution.

---

### Meta-Review · Area_Chair_dipp · 2024-12-20

**Metareview:**

This paper studies the role of depth in deep linear networks. In contrast to prior works, the paper argues that depth can speed up convergence. To demonstrate this, the paper examines a regression with quadratic loss and propose an algorithm (AGD)  with fast convergence and depth acceleration. The paper also provides empirical analysis to provide more evidence for this.

The reviews for the paper were mostly negative. The primary concerns of the reviewers were: (1) limited practical implications of the results in the paper (2) fairly contrived settings where the benefits of depth are shown. In particular, the paper mainly focuses on setting where width of the layer is 1. While I don't fully agree with the reviewers comments, I do agree that the setting studied in the paper looks somewhat contrived and is not very interesting from a practical viewpoint (even though I understand this is not the intent of the paper). I also think the paper can be revised to improve presentation and provide more practical implications of the paper. I recommend rejection in the current form.

**Additional Comments On Reviewer Discussion:**

The discussion mainly happened regarding the scope of the setting studied in the paper. While the authors clarified some of the misunderstandings of the reviewers, I think there are still valid concerns regarding the paper.

---

### Decision · Program_Chairs · 2025-01-22

Reject